# Expansion of the Invasive Plant Species *Reynoutria japonica* Houtt in the Upper Bistriţa Mountain River Basin with a Calculus on the Productive Potential of a Mountain Meadow

**Bogdan-Mihai Negrea** [1,2,*], **Valeriu Stoilov-Linu** [1,3], **Cristian-Emilian Pop** [4,5], **György Deák** [5,*], **Nicolae Crăciun** [6] and **Marius Mirodon Făgăraş** [2]

1   CE-MONT, Mountain Economy Center of the "Costin C. Kiritescu", National Institute of Economic Research—INCE, Romanian Academy, 49 Petreni Street, 725700 Vatra Dornei, Romania; linu_valeriu@yahoo.com
2   Doctoral School of Applied Sciences (Biology), Ovidius University of Constanta, 58 Ion Voda Street, 900525 Constanta, Romania; marius_fagaras@yahoo.com
3   Geography Department, Geography and Geology Faculty, Alexandru Ioan Cuza University of Iasi, 20A Carol I, 700505 Iasi, Romania
4   Non-Governmental Research Organization Biologic, 14 Schitului Str., 032044 Bucharest, Romania; pop.cristian-emilian@ngobiologic.com
5   National Institute for Research & Development in Environmental Protection, 060031 Bucharest, Romania
6   Zoology Section, Department of Biochemistry and Molecular Biology, Faculty of Biology, University of Bucharest, 91-95 Splaiul Independenţei Str., 050095 Bucharest, Romania; nicolai.craciun@bio.unibuc.ro
*   Correspondence: bogdannm@yahoo.com (B.-M.N.); dkrcontrol@yahoo.com (G.D.)

**Abstract:** Many invasive plant species use interactions with their anthropic environment as a propagation factor and benefit from climate changes, which have become accentuated in the last decade. The way such species interact with climate changes, as well as their high specific ecological plasticity, gives them a consistent advantage over native plant species. This work aims to demonstrate through a simple calculation the quantification of the productive potential of a wet meadow on which populations of an invasive plant species grew. The loss of productive potential induced by *Reynoutria japonica* Houtt on a mountain meadow in Ciocăneşti village, Romania, was the main objective. In the case of the productive potential of the meadows, a method for the general calculation of such losses was shown. The degree of anthropization of the studied area was also evaluated, correlating the degree of anthropization with the invasive species' potential for spreading and affecting the mountain area.

**Keywords:** losses quantification; Bistriţa River; meadows; non-native species; riparian habitats; *Reynoutria japonica* Houtt; meadow

## 1. Introduction

Invasive plant species outcompete native ones, causing damage to the ecosystem, and are recognized as a major threat to biodiversity. Among such species, *Reynoutria japonica* Houtt (Japanese knotweed), an herbal perennial plant from the Polygonaceae family [1] that is native to eastern Asia, can grow to heights of up to 3 m and has thick rhizomes, which can reach 4.5 m deep and up to 20 m$^2$ spread [2].

This fast-growing invasive plant is renowned for its traits, such as high adaptability and strong regeneration capacity, and also for the significant negative impact on its surroundings through secondary metabolites that inhibit native plants and even microbial communities [3].

In the current literature, there are only a few studies on the impact of such invasive species in the Romanian Bistriţa River basin; however, there are some studies that indicate the invasion and negative impact of the species in the mentioned area. The real damage to



the natural environment is very difficult to estimate because of multiple factors involved in the estimation, at least from an economic point of view [4,5]. Furthermore, most of the studies regarding this topic and region were performed more than a decade ago [6–13]. Therefore, we considered a survey of the mentioned area focused on *Reynoutria japonica* spread, which was not only necessary for the current evaluation of the mountain meadow's productive potential but will also be needed as a reference point for future surveys on the invasive potential of *Reynoutria japonica*.

In regard of the impact of non-native species, it is considered that any penetration of a new species only damages the environment, regardless of the major or minor ecological impact of the new species [8]. Many of these non-native plant species are particularly opportunistic and "occupy" areas of land quickly, in the areas where its spread was assisted by works of different types such as the construction of buildings, roads, hydro-technical structures, bridges, or railways. Many of these new plant species may have trophic and edaphic requirements usually associated with high climatic plasticity, much different or higher than that of native species, this is also the main reason why they adapt very well to the new conditions created by these superficial soil disturbances [6].

Many allochthone plant species form clonal populations/structures, which impress with their abundance, dominance, and massiveness (e.g., *Conyza canadensis*, *Reynoutria japonica*, *Impatiens glandulifera*, and *Robinia pseudoacacia*). Their impact on native species is physically observable and quantifiable by biomass assessments as well as the alteration of native habitats [4,5,14].

With the globalization process and the often-uncontrolled movement of unprocessed plant products, an invasion of opportunistic plants that often partially replace the native flora is taking place. This invasion is accelerated by the increasing rate of seasonal human migration of the local population, but mainly by the speed at which goods move globally, such as international imports of agricultural goods and other products via trains and trucks, imports that increased in the developing general area of Ciocănești and upper Bistrița River basin [4]. The aim of this work was to evaluate the state of the invasion of *Reynoutria japonica* Houtt in the surveyed area using an unmanned aerial vehicle (UAV) for spectral data gathering, coupled with mapping and classic assessment methods and also to quantify the productive potential of the area affected by this species.

## 2. Materials and Methods

The research location is represented by the upper part of the Bistrița River basin, which has a total length of 283 km and a basin surface of 7039 km$^2$ [15]. Because the basin surface is extremely large, the study was focused on the upper mountain course with a surface of 1170 km$^2$ (Figure 1). The compartment surveyed in the study was mostly an intramountain corridor with a narrow valley and a high depression area (average altitude is 800 m). This location was chosen because there are very little data regarding the invasiveness potential and productive potential on riparian artificial meadows situated on these mountainous river terraces. The study area, with a total surface of 1828.77 hectares (ha), has an average degree of anthropization, with the largest city being the city of Vatra Dornei, which has a population of 14,429 residents [16]. The climate is subarctic with cool summers and year-round rainfall, with the coldest month averaging below 0 °C and 3 months averaging above 10 °C, with no significant precipitation difference between seasons [17]. The location was established on a first terrace meadow, on the left bank of the Bistrița River, located on the territory of Ciocănești village administrative territorial unit (ATU), Suceava county, located on terrace 1 of the upper Bistrița River basin (Figure 1).

In order to identify and evaluate as accurately as possible the impact of the invasive plant populations in the study area, we used classical botanical study methods as well as phytocoenological modern evaluation methods using orthophoto images obtained with UAV and CORINE Land Cover maps.

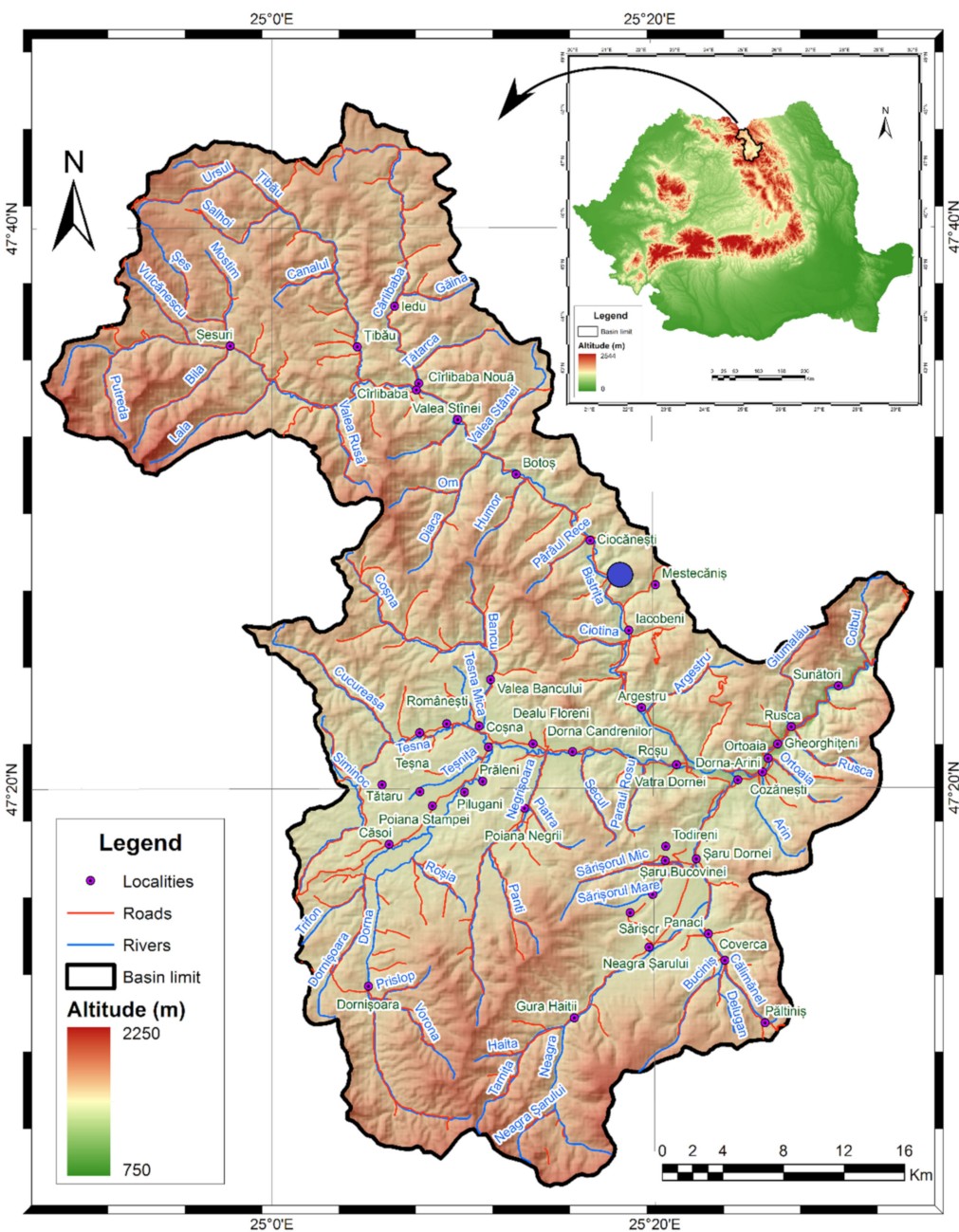

**Figure 1.** Upper Bistriţa River basin with localization of the investigated meadow (blue spot, Ciocăneşti).

## 2.1. CORINE Land Cover Assessment Method

CORINE Land Cover (CLC) maps help in obtaining environmental indicators and planning and managing natural resources, but at the same time, they can also support decision makers [18]. Moreover, the CLC dataset can be used not only to visualize terrain and measure terrain parameters but also to monitor landscapes over time [19]. To better assess the anthropogenic impact of the study area, a comparative study based on CLC database public data from the last 28 years (1990–2018) was used. CLC is the first operational program in the world to use satellite data under the auspices of the European Environment Agency [20].

Mapping to obtain the CLC dataset was performed using LANDSAT 5 MSS/TM satellites for the 1990 layer, LANDSAT 7 ETM for the 2000 dataset, SPOT-4/5 images, and IRS P6 LISS III in 2006, IRS P6 LISS III and RapidEye for those of 2012, and finally Sentinel

2 and LANDSAT 8 for the 2018 layer. This database provides useful information on how to cover/use the land and also has the benefit that it is available for a considerable period of time and can be an important starting point both in terms of land cover analysis and the impact of these changes on the environment [21].

The dynamics of the polygons assigned to the land use classes reflecting the anthropic expansion were analyzed (111, 112,121, 122, 131, 132, 133, 141, 142; CORINE Land Cover Classification). The CLC analysis involved the comparison of 5 consecutive CLC layers between 1990 and 2018, accessible from the Copernicus database (Figures 2 and 3). In addition, using the means and statistical data provided by CLC, the habitat fragmentation was analyzed.

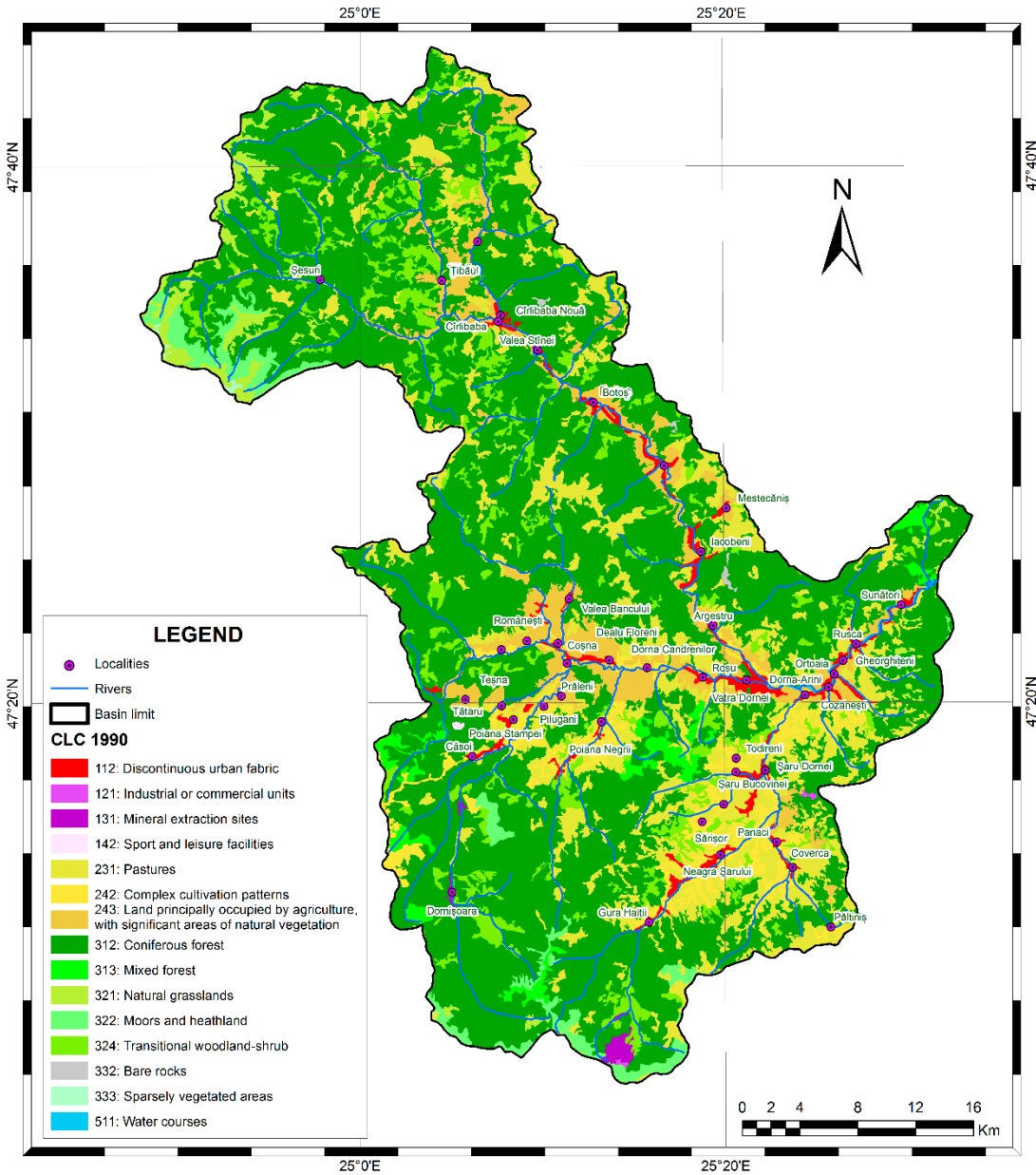

**Figure 2.** CORINE land cover 1990.

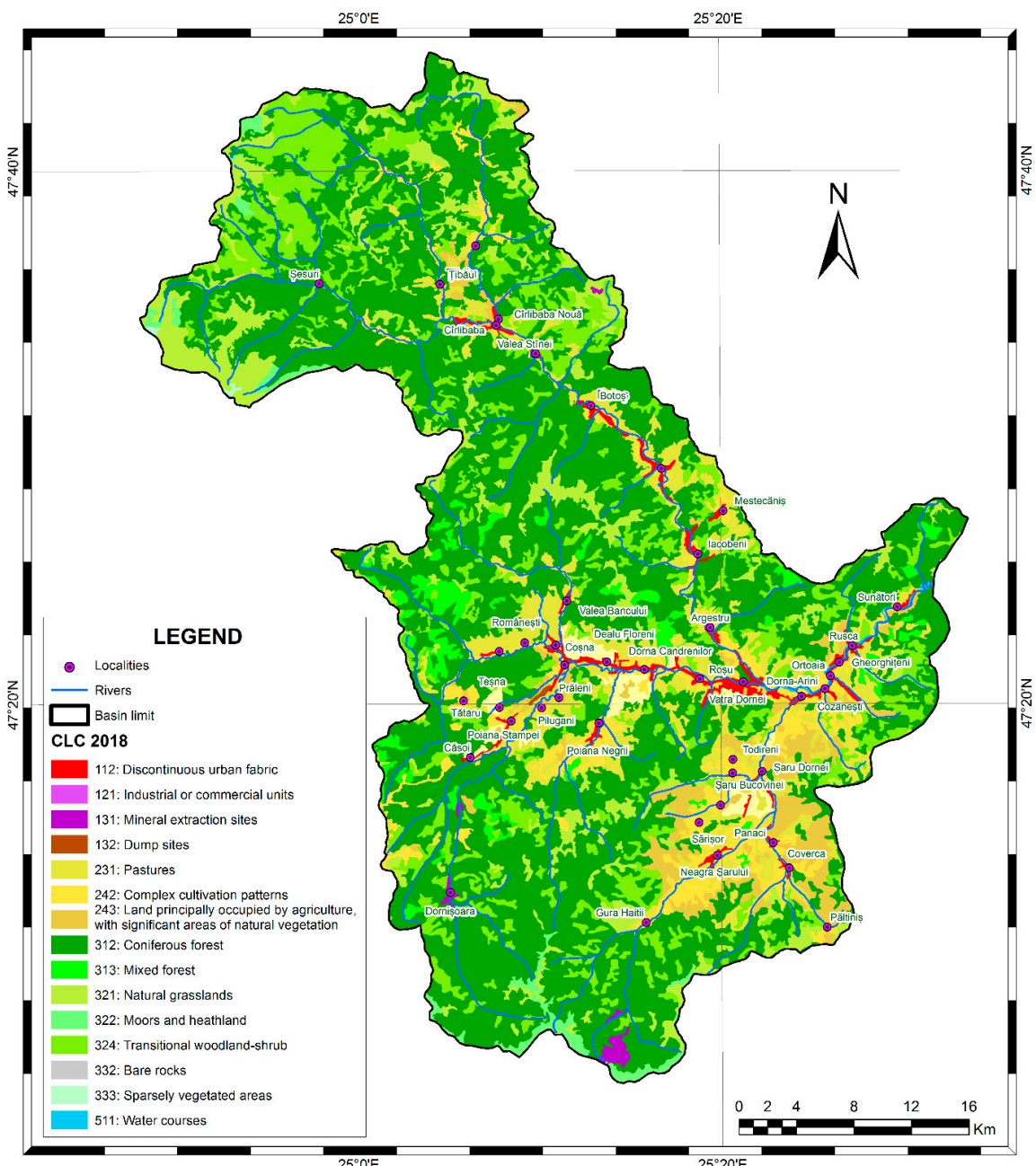

**Figure 3.** CORINE land cover 2018.

## 2.2. Classical Phytocoenological Methods

To characterize the analyzed meadows, a total of 9 survey areas were selected, and each sample area was chosen randomly. Spontaneous flora species were identified on each 1 m$^2$ sample area. The phytosociological research methods of the Central European School elaborated by Braun-Blanquet in 1932 [22] and adapted by Borza and Boşcaiu in 1965 [23] to the particularities of Romania were utilized. The technique of surveying and the use of quantitative and qualitative phytocoenological indices were in accordance with the indications [24].

The most homogeneous test surfaces were chosen to be randomly characteristic of phytocoenosis, depending on the nature and complexity of their horizontal and vertical structure. The analysis was carried out close to the full development of meadows with the mention that, in the year of the experimental work (2021), the maximum phenological development was delayed by

a relatively low average temperature at the beginning of the summer (≈10 °C), which led to a delay of maximum development by approximately 3 weeks.

The size of the test areas was 1 m$^2$; in these 9 test areas, the total identification of existing plant species and the total collection (up to ground level) of all biomass were performed. In the centralization table of the surveys (Table 1), the abundance or the dominance of the species was noted with values from 1 to 5, insignificant presence was noted with (+), and the total absence of the species from that survey/sample square was noted with (−). Subsequently, in situ and ex situ weighing of the samples of plant material thus obtained was performed to determine the total green/dry mass per sample (Figures S1–S5).

**Table 1.** Qualitative and quantitative composition expressed by the grouping of the surveys in the case of the analyzed meadow.

| No. sp. | Sample Code | P1 | P2 | P3 | P4 | P5 | P6 | P7 | P8 | P9 |
|---|---|---|---|---|---|---|---|---|---|---|
| | Species | | | | | | | | | |
| 1 | *Achillea millefolium* L. | - | - | + | | - | - | + | - | + |
| 2 | *Alchemilla vulgaris* L. | + | + | + | 2 | + | + | + | + | + |
| 3 | *Alopecurus pratensis* L. | 1 | 1 | 2 | 1 | 1 | 1 | 2 | 1 | 2 |
| 4 | *Anthoxanthum odoratum* L. | 1 | 1 | 1 | 1 | 1 | 1 | 1 | 1 | 1 |
| 5 | *Cardamine pratensis* L. | + | + | - | - | + | - | - | + | + |
| 6 | *Carex nigra* L. | + | + | + | + | - | + | + | + | - |
| 7 | *Campanula patula* L. | - | - | - | - | - | 1 | - | + | - |
| 8 | *Crepis mollis* (Jacq) Asch. | + | + | + | + | + | + | + | 1 | + |
| 9 | *Cruciata glabra* ssp. *glabra* L. | + | + | + | + | + | + | + | + | + |
| 10 | *Dactylis glomerata* L. | 2 | 2 | 3 | 3 | 3 | 3 | 3 | 3 | 3 |
| 11 | *Dechampsia cespitosa* (L.) P. Beauv. | 1 | 1 | 1 | 1 | 1 | 1 | 1 | 1 | 1 |
| 12 | *Festuca nigrescens* L. | + | + | 1 | + | + | 1 | + | + | + |
| 13 | *Festuca rubra* L. | + | 1 | + | + | + | + | + | 1 | 1 |
| 14 | *Hieracium aurantiacum* L. | + | - | - | - | + | - | - | + | - |
| 15 | *Hypericum perforatum* L. | + | + | - | + | - | - | - | + | - |
| 16 | *Leucanthemum vulgare* Lam. | 2 | 2 | 2 | 2 | 2 | 2 | 2 | 2 | 2 |
| 17 | *Lotus corniculatus* L. | + | + | + | + | + | - | + | + | - |
| 18 | *Lychnis flos-cuculi* L. | 1 | 1 | 1 | 2 | 1 | 1 | 1 | 1 | 1 |
| 19 | *Myosotis scorpioides* L. | + | + | + | + | + | 2 | + | + | + |
| 20 | *Nardus stricta* L. | + | | + | + | - | + | - | + | - |
| 21 | *Phleum pretense* L. | 1 | 1 | 1 | 1 | 1 | 1 | 1 | 1 | 1 |
| 22 | *Pimpinella major* (L.) Huds. | + | + | - | - | - | + | - | + | - |
| 23 | *Plantago lanceolata* L. | 1 | 1 | + | 1 | + | 1 | + | 1 | + |
| 24 | *Plantago media* L. | - | + | - | - | 1 | + | - | + | - |
| 25 | *Poa anua* L. | 2 | 1 | 2 | 2 | 1 | 2 | 2 | 1 | 2 |
| 26 | *Poa angustifolia* L. | 1 | 1 | 2 | 3 | 3 | 3 | 2 | 2 | 3 |
| 27 | *Poa trivialis* L. | + | + | 1 | 1 | 1 | 1 | 1 | 1 | 1 |
| 28 | *Poa palustris* L. | 1 | 1 | + | 1 | 1 | + | 1 | + | 1 |
| 29 | *Poa pratensis* L. | 2 | 1 | 4 | 3 | 3 | 3 | 3 | 3 | 3 |
| 30 | *Poa pratensis* ssp. *pratensis* L. | 2 | 1 | 2 | 2 | 1 | 3 | 1 | 2 | 2 |
| 31 | *Polygonum bistorta* L. | - | - | 1 | 3 | 2 | - | 1 | 2 | - |
| 32 | *Potentilla erecta* (L.) Raeusch. | + | + | + | + | + | + | + | + | + |
| 33 | *Prenanthes purpurea* L. | - | + | - | - | - | + | - | + | - |
| 34 | *Ranunculus acris* ssp. *acris* L. | 2 | 2 | 2 | 2 | 2 | 4 | 2 | 2 | 2 |
| 35 | *Ranunculus repens* L. | + | + | + | 1 | + | + | + | + | + |
| 36 | *Rhinanthus minor* L. | 1 | 1 | 1 | 1 | 2 | 2 | 1 | 1 | 1 |
| 37 | *Rumex acetosa* L. | 2 | 2 | 3 | 2 | 2 | 2 | 2 | 2 | 2 |
| 38 | *Scabiosa columbaria* L. | 1 | + | + | + | + | + | + | + | + |
| 39 | *Stellaria graminea* L. | 1 | 1 | 1 | 1 | 1 | 1 | 1 | 1 | 1 |
| 40 | *Taraxacum officinale* Weber s. L. | 2 | 3 | 2 | 3 | 2 | 2 | 2 | 3 | 2 |
| 41 | *Tragopogon pratensis* L. | - | + | - | - | + | - | - | + | + |
| 42 | *Trifolium hybridum* L. | 1 | 1 | 1 | 1 | 1 | 1 | 1 | 1 | 1 |
| 43 | *Trifolium pretense* L. | + | 1 | + | 1 | + | + | + | 1 | + |
| 44 | *Trifolium repens* L. | 1 | 1 | 1 | 1 | 1 | 1 | 1 | 1 | 1 |
| 45 | *Veronica chamaedrys* L. | + | + | + | + | + | + | + | + | + |

P1–P9 represent the survey plots; (+) present species; (-) absent species.

The quantitative indices approached was used to calculate abundance, and the Braun-Blanquet [25] method was used for the evaluation of species coverage, for which actual coverage (projection) and absolute coverage (basal) were taken into account. The inventory

periods were June–July 2021, and ArcMap 10.8 (ArcGIS) from ESRI, QGIS 3.12, Microsoft Office (Word, Excel), and Inkscape 1.02 software were used for data processing.

### 2.3. Spectral UAV Scanning Methods

To obtain qualitative information, it is necessary to use a UAV in a controlled handling system. This mode of control of the drone involved prior programming of the equipment establishing the flight path, altitude, flight speed, and frequency of photographic recordings. Based on these settings and with known spatial coordinates, high-resolution orthophoto images were obtained with the use of the UAV, and then the images were processed for quantitative measurements and evaluations.

The delimitation of the flight area was performed interactively within the program, with the support of satellite images from different sources (Google Earth and Open Street Map). The successive coverage of the frames taken by the UAV was at least 80%. The sensor used was a multispectral (Sequoia) type.

### 2.4. Inverse Distance Weighted (IDW) Interpolation Method

For the estimation of the expansion possibility of *R. japonica* Houtt in the meadows of the upper Bistrița basin, we used the inverse distance weighted (IDW) [26] interpolation method. IDW interpolation determines cell values using a linearly weighted combination of a set of sample points. The weight is a function of inverse distance, the surface being interpolated should be that of a locational-dependent variable. IDW uses the measured values surrounding the prediction location to predict a value for any unsampled location, based on the assumption that things that are close to one another are more alike than those that are farther apart (ESRI ArcMap 10.8 Help Manual, 2020).

## 3. Results

Altimetric analysis represented by the digital terrain model (DTM) of the sampling area reveals that the relief energy is low, because the location corresponds to the first terrace of the major bed of the Bistrița River, with only a 4 m level difference over the entire investigated surface, between maximum and minimum values of 860–856 m alt. (Table 2).

**Table 2.** Centralization of the analyzed indices.

| DTM | NDVI | NDRE | GNDVI | GRVI |
|---|---|---|---|---|
| min: 856—max: 860 m | min: −0.40—max: 0.90 | min: −0.70—max: 0.58 | min: −0.43—max: 0.85 | min: 0.38—max: 12.53 |

### 3.1. Spectral UAV Scanning Results

The normalized difference vegetation index (NDVI) is a non-linear processing of the visible band RED (red) and NIR (near infrared), which is defined as the ratio of the difference between these bands and their sum. NDVI, which can be considered not only as an indicator of vegetation development but also of its density, is associated with parameters such as biomass, the share of vegetation cover, or photosynthetic activity. The NDVI analysis highlights that the investigated area has maximum NDVI values of over 0.90, while the lower value presented was −0.40. The NDVI spectral response reflects, in the case of all analyzed lots, an increased presence of healthy vegetation (Table 2; Figure S8).

The normalized difference red edge index (NDRE) is a metric value used to analyze whether or not multispectral images contain healthy vegetation, similar to NDVI, but using the ratio of the near-infrared (NIR) to the red edge band (REDEDGE). Compared to NDVI, NDRE is a more effective indicator of plant conditions for mid- and late-season crops, which have already accumulated a large amount of chlorophyll, because the REDEDGE band has a greater ability to penetrate through plant elements, compared to the RED band [27]. According to this index, the investigated area expressed an index of −0.82, which reflects a relatively worrying situation. This can be based on the submaximal phenological stage of the meadow (the meadow has not reached its maximum development) (Table 2; Figure S8).

The green normalized difference vegetation index (GNDVI) is similar to NDVI, except that instead of the red spectrum it measures the green spectrum in the range of 0.54 to 0.57 μm. This is an indicator of photosynthetic activity and implicitly reveals the level of vegetation cover. It is most often used to evaluate the moisture content and nitrogen concentration of plant leaves according to multispectral data that do not have an extreme red channel.

Compared to the NDVI index, it is more sensitive to chlorophyll concentration and is used in assessing aging and declining vegetation [28]. In the investigated surface, it varied between −0.43 and 0.85, fact-based on the inhomogeneity of the surface (Table 2; Figure S8). The vegetation ratio of the green ratio (GRVI) is a spectral indicator of vegetation, calculated based on the values of the visible reflectance, GREEN (green) and RED (red). The GRVI method uses the GRVI time series instead of NDVI and determines the maximum time of vegetation as well as its autumn decay coloration [29]. In the investigated surface, it varied between 0.38 and 12.53, based on the inhomogeneity of the surface, (Table 2; Figure S8).

### 3.2. Floristic Characterization and Evaluation of the Analyzed Meadow's Productivity

The surveyed area was a wet hayfield (soils moist and rich in raw humus), with *Poa pratensis* L. (Table 1). Co-dominant and dominant species, subtype *Poa pratensis* thrives on humus-rich, reclaimed moist soils in medium-sized areas with high water reserve. It has a high feed value (VF8), with a protein content of up to 13% crude protein [30]. The meadow has no conservative value and is part of habitats of the type (R3716)—Danubian-Pontic meadows with *Poa pratensis*, *Festuca pratensis*, and *Alopecurus pratensis*.

According to the local agriculturists, fertilization is carried out with medium amounts of organic fertilizer (10–30 t/ha), with a maintenance regime of semi-intensive mowing and extensive grazing. The meadow is mowed or grazed semi-extensively. Therefore, the meadow has a medium–high anthropogenic influence due to a particular maintenance regime. The productivity of green mass calculated by mediation on the nine samples collected related to crop 1 was 25.3 t/ha/year.

### 3.3. CORINE Land Cover Analysis Results

The CORINE Land Cover analysis performed in the upper Bistriţa River basin (Figure 2 dated from 1990 and Figure 3 dated from 2018) revealed a rapid anthropization of the area. Thus, natural areas lost up to 3% of their original habitats, or major changes occurred in them (buildings, roads, utility networks). In the case of agricultural areas, the largest reduction occurred in the case of pastures (231) from 14.8% coverage in 1990 to 8.6%, in 2018, means no less than 15,727 ha, of which we estimate that at least 5%, that is, 786.3 ha, were affected by invasive species, especially meadows on the banks of the river Bistriţa and its tributaries, with the other categories remaining relatively constant. The visual analysis of the spatial maps shows an increase in the degree of fragmentation of land cover types, related to two main factors: the quality and high resolution of satellite images used in recent years that allows a better interpretation of land cover, respectively, changes that took place between the categories of forests and areas of transition between forests and shrub vegetation. Overall, the area is quite well preserved in terms of fragmentation with minor implications on habitat degradation. Degradation is more visible only in areas with an industrial impact (mining), quarries, tailing dumps, etc. However, the built-up areas in the analyzed area showed a tendency to double in size compared to the 1990s, which is largely due to the expansion and development of tourism in the area by expanding in particular accommodation and tourism facilities. In the recent past years, the anthropical impact was amplified and modified the surfaces of the riparian habitats with natural or semi-natural meadows.

### 3.4. Inverse Distance Weighted (IDW) Interpolation Results

The inverse distance weighted (IDW) interpolation method was applied to a number of known locations in the upper Bistriţa River basin where populations of the *R. japonica*

Houtt species are currently known to expand. The interpolation result is a risk map shown in Figure 4, which reflects the possibility of the spread of the species across the meadows of the entire upper basin, including the vicinity of the sampled plot.

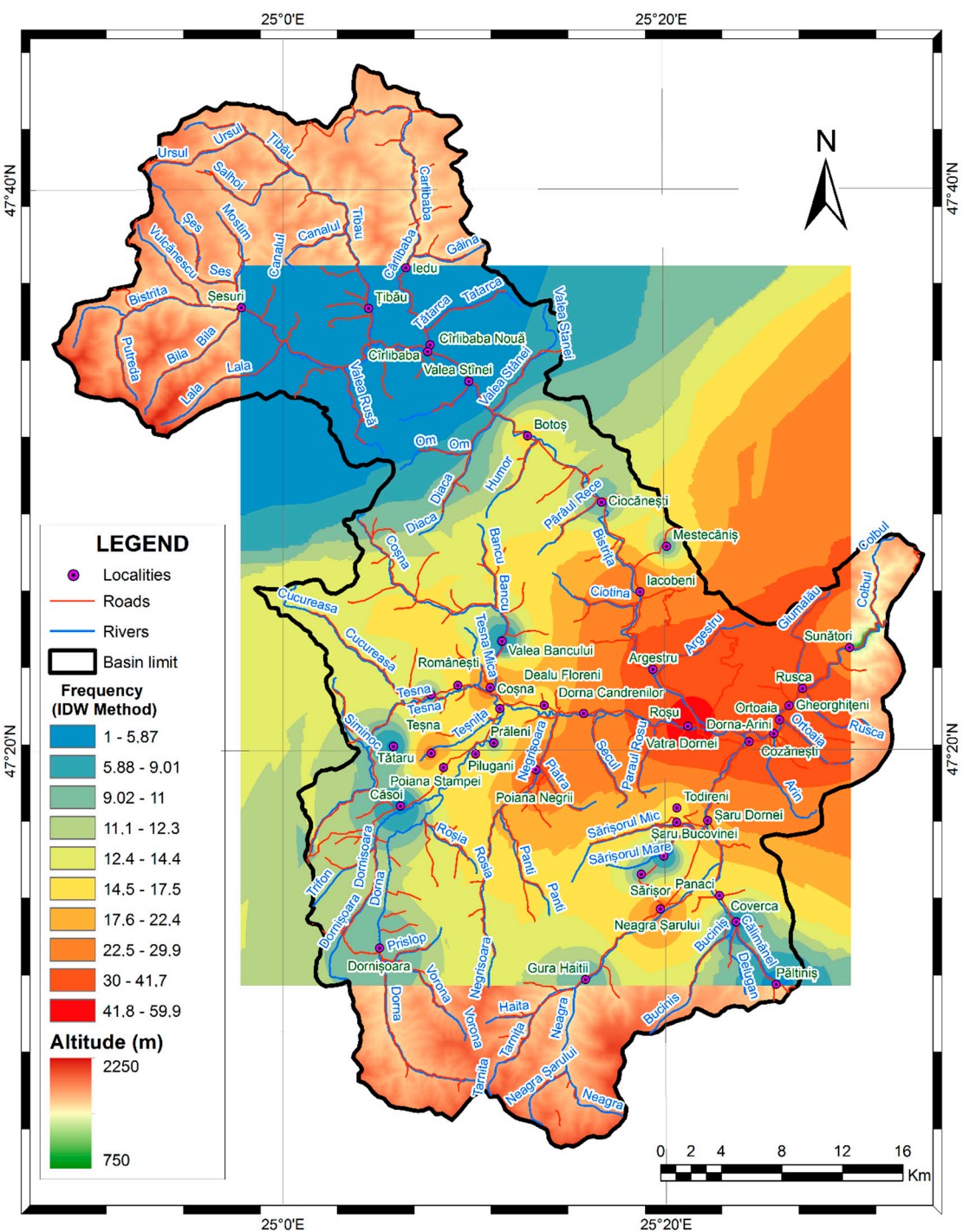

**Figure 4.** The IDW map of *R. japonica* Houtt populations in upper Bistrița River basin.

## 4. Discussion

Habitat loss through degradation and fragmentation is considered an important cause of species extinction, with the main cause currently being human activity. The habitat of a species refers to the specific "place" where that species is found. All species have a specific

need for habitat, the more specific these needs are, the rarer and more localized the habitat is; therefore, the possibility of losing such habitat is increased [31].

### 4.1. Habitat Degradation

The factors that lead to habitat degradation in the analyzed meadows were analyzed, taking into account the impact of fragmentation and anthropization on natural grassland habitats. At the same time, a calculation was carried out on the production of these artificial wet meadows, from which it was possible to calculate a possible loss of potential due to the spread of the invasive species *R. japonica* Houtt. Among the most important factors that lead to the degradation of ecosystems, in addition to overexploitation of resources, pollution, and contamination, invasive species that enter the ecosystem and produce imbalances by occupying ecological niches were highlighted. With the overexploitation of meadows, their ruderalization occurs, and thus, the appearance and multiplication of invasive plant species that animals do not consume, such as *R. japonica* Houtt, takes place.

Among the invasive plant species that most severely degraded the habitat and the productivity of the surveyed grasslands in the meadows of the upper Bistrița River basin, our research focused on *R. japonica* Houtt (Figures S6 and S7), the most aggressive species in the area. Habitat degradation is referred to as modifications or alterations of an ecosystem in the sense of decreasing its quality [31].

As a native species from East Asia, *R. japonica* Houtt was introduced in Europe to be used as an ornamental plant [32]. Gradually, the species became known throughout the continent as an adventitious plant, decorative in shape and color. The species is very plastic from an ecological and climatical point of view, it was introduced in Romania in parks and botanical/private gardens from where it got out of control, being declared sub-spontaneous and later invasive. In the year 2021, over 60 populations were registered across the Bistrița River basin [5]. The species has been classified as "highly invasive" by the CABI Invasive Species Compendium [33].

*R. japonica* Houtt invades semi-natural mountain meadows with a high to average productivity, as in the case of the one chosen in our survey as an example of calculation. Currently, the populations of this species are out of control and occupy considerable areas on the riverbanks and the major riverbeds of some tributaries in the upper basin of the Bistrița River. The colonies have areas between 0.5 and 100 $m^2$ and have the ability to completely replace indigenous plant communities in large areas by shading and also through their secondary metabolites that inhibit native plants [3]. It also has the ability to maintain a certain resource balance in the habitats it creates by changing the availability of resources for native plants and access to light; therefore, native species are unlikely to compete with them. The dense, and high (up to 3 m), plant communities composed by this species (Figures S6 and S7), shade the soil, reducing by up to 90% the incident solar radiation, confirmed through measuring the incident light.

Climate change though global warming and anthropization, which have become evident in the last decade, may become the main driving factors for accelerating its spread within the Bistrița River basin. Due to these climatic phenomena, native plant species especially suffer a physiological pressure. These climate changes in recent years are inducing the degradation of native habitats to the advantage of aggressive non-native species, which tend to adapt much better to changes in ecological niches, because they show a much higher ecological plasticity than native species. An important aspect worth mentioning is that although having a negative environmental impact, some invasive species (e.g., *Lonicera japonica*) can be very useful in green building technologies, especially in hot and humid subtropical climates where strong solar radiation imposes heat stress upon building users [34,35].

*R. japonica* Houtt occupies more and more territory in the upper Bistrița River basin; the number but also the size of the studied colonies increases from year to year. This phenomenon occurs through the occupation of the useful land, limiting or even stopping the useful productivity of native plants that are part of the natural composition of the wet

meadows. The pressure exerted on the native habitats is of great importance, because, in the upper Bistrița River basin mostly located in the mountain area, useful lands are a limited resource.

*4.2. The Calculation of the Impact*

The calculation of the impact that this species may have on the productive potential of the meadows of the upper basin of the Bistrița can be easily reflected by the following reasoning: If we estimate a percentage of 15% (mean) due to the lack of uniformity of vegetation on the land surface, we obtain a total of 21.5 t/ha total productive potential of green biomass at first mowing on these meadow types; if from this productive potential of 21.5 t/ha we decrease a percentage of at least 10–15% due to the coverage with the invasive species *R. japonica* Houtt, practically the production of useful green biomass decreases by the same percentage, meaning 15%, resulting in a loss of productive potential of 3.2 t/ha.

This amount of biomass produced by *R. japonica* Houtt cannot be used as fodder, as it means, at most a short/medium-term carbon sequestration. We call it short-term carbon sequestration because the exposed living biomass decomposes rapidly in 2–3 years without creating a significant turbic or humic deposit, due to a weak presence of plant fiber in the composition of the aerial parts of this species.

This invasive aspect can become a major inconvenience if it is taken into account that almost all the meadows in the upper Bistrița basin are areas with a major risk of spreading for this species (Figure 4), as they are situated in the immediate vicinity of the river and its tributaries.

In the meadows of Vișeu River, Maramureș County, a percentage of up to 50–55% coverage with *R. japonica* Houtt was assayed [14]. A concerning fact being that this area is considered to have similar mountainous ecological conditions as the Bistrița River valley.

If taken into account that the expansion of the *R. japonica* Houtt is at its beginnings in these riparian mountain areas, and because the riparian meadows can be, and often are, covered in much larger proportions, we can anticipate the magnitude of a major future disturbance and the uncontrolled spread of this species on large areas (meadows) through all the superior basin of Bistrița River.

**5. Conclusions**

From the point of view of the CORINE Land Cover analysis, the analyzed area has lost its natural habitat, especially in the vicinity of manmade areas. Natural habitats and meadows have been modified by human intervention by up to 3%. This percentage does not seem high; however, considering that this is a mountainous area with limited usable land (in terms of agriculture area) that is situated in the vicinity of natural and artificial meadows, it can be observed that much of this land has been affected by human interventions and is, thus, prone to the invasive alien species *R. japonica* Houtt.

Although slightly inhomogeneous, the standard sample area taken into account can be characterized as a fertile and healthy meadow whose productivity can be a basis for calculating the productive potential of riverside meadows in the mountains of the upper basin of Bistrița River. The stand reflects ideal conditions of productivity and vegetation. This was demonstrated by taking into account the productivity and health (vegetation indices) characteristics of the analyzed meadow.

The inverse distance weighted (IDW) interpolation result is a risk map shown in Figure 4, which reflects the possibility of the species to spread across the meadows of the entire upper basin, including the vicinity of the sampled plot. The only potentially limiting factor of this species is the existence of colder local microclimates, determined by altitude.

In making this brief calculation of the loss on productive potential, we would like to draw attention to the phenomenon of the spread of *R. japonica* Houtt in the riparian meadows in the mountainous area of the upper Bistrița River basin and how the populations of this species may affect a significant percentage of the productivity of the meadows (a minimum 15%) located in this intramountain territory.

**Supplementary Materials:** The following supporting information can be downloaded at https://www.mdpi.com/article/10.3390/su14095737/s1: Figures S1–S7: captures from the field campaign (2021), Figure S8. DEM; NDVI; INDRE; GNDVI; GRVI, and Red, Green, NIR, and RE Reflectance.

**Author Contributions:** Conceptualization, B.-M.N. and V.S.-L.; methodology, B.-M.N. and V.S.-L.; software, V.S.-L.; validation, B.-M.N. and V.S.-L.; formal analysis, B.-M.N. and V.S.-L.; investigation, B.-M.N. and V.S.-L.; resources, C.-E.P. and N.C.; data curation, B.-M.N.; writing—original draft preparation, B.-M.N.; writing—review and editing, C.-E.P.; visualization, N.C. and M.M.F.; supervision, G.D. and M.M.F.; project administration, B.-M.N.; funding acquisition, C.-E.P. and N.C. All authors have read and agreed to the published version of the manuscript.

**Funding:** This research was partially funded by ADER, grant number 16.3.4., and also through private resources.

**Institutional Review Board Statement:** Not applicable.

**Informed Consent Statement:** Not applicable.

**Data Availability Statement:** Not applicable.

**Acknowledgments:** We would like to thank our colleagues Ionel Popa, Sabin Nicula and Surdu Ioan for their encouragement, ideas and guidance so that these results could be published. We also want to thank to the Non-Governmental Research Organization Biologic for their ongoing support.

**Conflicts of Interest:** The authors declare no conflict of interest.

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
