# Peer review of "Expansion of the Invasive Plant Species Reynoutria japonica Houtt in the Upper Bistrița Mountain River Basin with a Calculus on the Productive Potential of a Mountain Meadow"

_sustainability, doi:10.3390/su14095737_

Round 1
Reviewer 1 Report
The manuscript entitled "Expansion of the invasive plant species Reinoutria japonica 2 Houtt in the upper Bistrița River basin; a calculus on the loss of 3 productive potential of a mountain meadow" is not scientifically presented.
The authors have developed several analysis poorly introduced and without any questions. The general aims can only be imagined reading the title. Despite, the literature on invasive alien species, the introduction is too short. It's very hard to review a paper without any clue of what the authors want to investigate.
Author Response
Dear Reviewer,
Thank you for your remarks,
Please find attached our replies.
Best regards,
The authors

Reviewer 2 Report
This manuscript revolves around the problem of the spread of Japanese knotweed in the Upper Bistrița region. The significance of the research is unquestionable. The study area is a mountainous zone with minimal anthropogenic impact and minimal agricultural activity. When I first visited the area around 1990, there were practically no invasive plant species found, but that later changed, and in 2006 I also encountered Japanese knotweed in the area.
As the spread of invasive plants has already become a problem in relatively protected mountainous areas, the present manuscript may assist in the design of conservation treatments.
The introduction is well done, the research design appropriate. The description of the methods is also well done. For me the presentation of the results and their discussion is a little bit chaotic, I suggest for the authors to rewrite it to be more simple and logical in order to better understanding.
The name of the plant “Reinoutria” must be corrected to ‘Reynoutria’ in the title.
The potential readers of this work seems to be the water and nature conservation practitioners.
After the reviews mentioned above the manuscript will be suitable for publication.
Author Response
Dear Reviewer,
Thank you for your suggestions and remarks.
Please find attached our answer.
Best wishes,
The authors

Reviewer 3 Report
Abstract
The Abstract is well-written. With only minor polishing, the main points of the research are effectively articulated.
Line 22: These two terms, "non-native" and "invasive", can bear different meanings. In the manuscript, both terms can be found. Therefore, it is recommended that the authors reach a consensus which term to use. Better still, a definition of the chosen term can be given in the Abstract or the Introduction section. A search and replace may be helpful.
Line 22 to 33: There is no need of splitting the Abstract into different paragraphs. The authors should combine the paragraphs into a single text. This is an easy change.
Line 29: The authors should specify where the examined mountain meadow was. Perhaps the authors can just insert the name of the region or country, Romania.
Introduction
The Introduction section is sharply focused on the subject of the study. Some details should be provided. Please read the detailed comments.
Line 44 to 47: Among horticulturalist and plant scientists, it is well-known that Japanese knotweed can wreck a havoc in the ecology of many habitats. However, some readers may not be knowledgeable about this issue. It is suggested that the authors can add a previous case-study, no matter in Romanian context or not, after Line 47. Just cite a study, and describe the case in two to three sentences.
Line 66: Remove "we" and convert the sentence into passive voice. This can make the statement more objective.
Line 67 to 68: As the movement of people and goods may help speed up the spread of Japanese Knotweed, the authors are advised to provide some information about the international logistics and shipping in the study area. Just one or two sentences will do.
Line 70: The authors should place the aim of the study here. Although the aim was stated in the Abstract, readers still expect to read the aim of the study near the end of the Introduction section. Just copy the aim from the Abstract and paste it here.
Materials and methods
The methods used in this study are robust. Some details may be missing. But after further clarifications and specifications, this section will be ready for publication.
Line 78: The authors should specify where the examined mountain meadow was. Perhaps the authors can just insert the name of the region or country, Romania. This comment also applies to the Abstract.
Line 93 to 94: Climate and weather have an important role in the spread of a non-native or invasive species. The authors can find the climate type of the study area according to the Köppen-Geiger climate classification. This piece of information can be revealed by a quick search. Add a sentence to specify the climate type here
Line 120 to 122: Three methods are mentioned here, namely "classical", "phytocoenological", and "botanical". However, the subsequent headings of the sub-sections did not match these descriptors. Perhaps the authors can modify the headings. With a quick change, the consistency of the text can be improved.
Line 125: Is each survey area 1 m2 in size? Please clarify.
Line 147: If the study was conducted in summer, please specify the temperature and rainfall pattern in the summer of the study area. Adding one to two sentences will be good.
Results
The results are clearly organised. Some variables which are not mentioned in the Materials and Methods section pop up here. Minor editing can solve the problem.
Line 178 to 185: Some environmental parameters such as soil moisture and fertility were mentioned in this paragraph. However, these variables were not mentioned in the previous section. Readers may be confused. It is suggested that the authors can briefly supplement the relevant information in the Materials and Methods section.
Line 187: In Table 1, do "P1", "P2", etc mean the different survey plots? If yes, state the meaning of the abbreviations in the caption of the table.
Line 198 to 228: Different measures related to imagery were used here. Please introduce in the Introduction or Materials and Methods sections. Readers who are not familiar with remote sensing may be perplexed by these paragraphs. Just add some sentences in the appropriate section.
Line 230 to 232: The presentation of Table 2 seems to contain some formatting errors. Please amend if needed.
Line 244: How much is the lost areas contain Japanese Knotweed? Please specify.
Discussion
The contents in Discussion section pertain to the results. Some extra information can be specified.
Line 290: Remove "we" and convert the sentence into passive voice. This can make the statement more objective.
Line 292 to 299: Was the study area close to the land uses mentioned in this paragraph?
Line 311 to 317: In the opinions of the readers, to what extent did climate change influence the invasion of Japanese Knotweed?
Line 318 to 324: Japanese Knotweed is surely invasive. As the authors mentioned that this species was used for landscape planting. The use of building-integrated greenery has gained increasing popularity. Some species, e.g. Lonicera japonica, may be invasive yet very useful in vertical greening. The authors can add this point to the Discussion with reference to existing studies. These two sources showcase the use of potentially invasive yet useful plant species for green buildings. Please cite these two sources:
- Lee, L. S., & Jim, C. Y. (2020). Multidimensional analysis of temporal and layered microclimatic behavior of subtropical climber green walls in summer. Urban Ecosystems, 23(2), 389-402.
- Lee, L. S., & Jim, C. Y. (2021). Quantitative approximation of shading-induced cooling by climber green wall based on multiple-iterative radiation pathways. In Eco-efficient Materials for Reducing Cooling Needs in Buildings and Construction (pp. 79-100). Woodhead Publishing.
Conclusion
There are no critical comments on this section.
Author Response
Dear Reviewer,
Thank you for your review report.
Please find attached our point-by-point replies.
Kind regards,
The authors

Round 2
Reviewer 1 Report
The manuscript has been deeply reviewed and I do appreciate the improvements done.
However, i suggest to improve the introduction which just introduce very shortly the topic of the paper.
I would end the introduction with the most relevant aims corrisponding with methodology adopted.
As I mention in the previous review even the discussion needs to be implemented because different paragraph introduce something going nowhere.
Author Response
Dear Reviewer,
Please find attached our replies to your suggestions.
Best regards,
The authors
